# Bio-Based Hydrogel and Aerogel Composites Prepared by Combining Cellulose Solutions and Waterborne Polyurethane

**DOI:** 10.3390/polym14010204

**Published:** 2022-01-05

**Authors:** Ling-Jie Huang, Wen-Jau Lee, Yi-Chun Chen

**Affiliations:** Department of Forestry, National Chung-Hsing University, 145 Xingda Rd., South Dist., Taichung City 402, Taiwan; a0975165765@gmail.com (L.-J.H.); wjlee@dragon.nchu.edu.tw (W.-J.L.)

**Keywords:** aerogel, cellulose, composite gel, hydrogel, waterborne polyurethane

## Abstract

Hydrogel composites can be prepared from cellulose-based materials and other gel materials, thus combining the advantages of both kinds of material. The aerogel, porous material formed after removing the water in the hydrogel, can maintain the network structure. Hydrogel and aerogel have high application potential. However, low mechanical strength and weight loss of cellulose hydrogel due to the water dehydration/absorption limit the feasibility of repeated use. In this study, cellulose hydrogels were prepared using microcrystalline cellulose (MC), carboxymethyl cellulose (CMC), and hydroxyethyl cellulose (HEC) as raw materials. Waterborne polyurethane (WPU) was added during the preparation process to form cellulose/WPU composite hydrogel and aerogel. The influence of the cellulose type and WPU addition ratio on the performance of hydrogel and aerogel were investigated. The results show that the introduction of WPU can help strengthen and stabilize the structure of cellulose hydrogel, reduce weight loss caused by water absorption and dehydration, and improve its reusability. The mixing of cellulose and WPU at a weight ratio of 90/10 is the best ratio to make the cellulose/WPU composite aerogel with the highest water swelling capacity and heat resistance.

## 1. Introduction

Hydrogel is a polymeric network material with low crosslink density, which has swelling ability and retains a large amount of water in the structure but is not soluble in water or solvents [1,2,3]. Hydrogels can be formed into lightweight, highly porous aerogels by appropriate dehydration methods while maintaining their network structure. This aerogel can absorb liquid several times its weight [4,5,6]. Cellulose is the most abundant renewable resource on Earth [7]. It has the characteristics of high toughness, biocompatibility, and biodegradability. The hydrogels and aerogels prepared from cellulose could combine their advantages. There have been many studies related to the application of cellulose hydrogels and aerogels, including controlled drug release [8,9,10], water and oil absorbents [11,12,13,14], and heavy metal adsorption [15,16,17].

Cellulose is highly hydrophilic, and the pores within the network structure can hold a large amount of water. The cellulose aerogel formed by dehydration of cellulose hydrogel has the potential to allow repeated water absorption and dehydration [18,19]. Our preliminary experiments found the characteristic of repeated morphology conversion between cellulose hydrogel and aerogel, which can facilitate the application of low transportation weight and high absorption capacity. However, the cellulose gel experiences weight loss during the repeated water absorption/dehydration process. In addition, cellulose hydrogels have the disadvantage of insufficient mechanical strength [20,21,22]. To improve this property and increase the scope of application, the introduction of natural or synthetic polymers into cellulose hydrogels or aerogels is a current research trend.

Liang et al. (2015) dissolved microcrystalline cellulose (MC) in 1-butyl-3-methylimidazolium chloride ionic solution and used *N*,*N*’-methylene bisacrylamide as a crosslinking agent to obtain MC hydrogel. This cellulose hydrogel was immersed in a mixed aqueous solution of ferric trichloride and sodium p-toluenesulfonate, and finally immersed in the pyrrole solution to form a semi-interpenetrating cellulose/polypyrrole hybrid hydrogel. The results indicate that the hydrogel has conductivity and can improve the mechanical strength of cellulose hydrogel [23]. Godiya et al. (2019) dissolved carboxymethyl cellulose (CMC) in water and mixed it with acrylamide (AM). They used *N*,*N*’-methylene-bis-acrylamide as a crosslinking agent and ammonium persulfate as an initiator to prepare CMC/AM hybrid hydrogel. The aerogel was obtained by freeze-drying of hydrogel. They produced a gel material with better mechanical properties and ability to adsorb heavy metals [24]. Wang et al. (2011) dissolved hydroxyethyl cellulose (HEC) in water and used ammonium persulfate as an initiator to generate free radicals in the cellulose solution. Then, NaOH-neutralized acrylic acid (AA), a medicinal stone (MS), and a MBA crosslinking agent were added. This method can obtain HEC/MS/AA hybrid hydrogels with high water absorption capacity and pH- and saline-sensitive properties [25]. Tang et al. (2020) added poly (propylene glycol adipate) (PPA) to a cellulose solution to prepare cellulose hydrogel and cellulose aerogel with a direct mixing method. The results suggested that using PPA as a modifier can effectively improve the mechanical strength of cellulose aerogel and has great potential in the treatment of printing and dyeing wastewater [22]. Verdolotti et al. (2019) immersed cellulose aerogel in hexamethylene-1,6 diisocyanate/acetone solution. They found a covalent bond between the isocyanate and the cellulose of the aerogel. This polyurethane (PU) polymer coating can further stabilize the network and strength of the aerogel, making it more suitable for various applications [26].

PU is a resin with a urethane structure obtained by reacting the NCO group of an isocyanate with the OH group of a polyol. Due to its good toughness, ductility, strength, wear resistance, and weather resistance, PU resin is widely used in adhesives, coatings, printings, biomedical polymers, architectural engineering, the automobile industry, textile processing, and other fields [27,28]. However, traditional PU resin involves a solvent-based system. A large amount of volatile organic compounds is released during the manufacturing and use processes, causing great harm to the biological environment. In recent years, due to the rising awareness of environmental protection, the development and utilization of waterborne polyurethane (WPU) have attracted much attention. In the synthesis process of WPU resin, hydrophilic groups are introduced to modify the hydrophobic PU main chain, and then water and a chain extender are added to carry out emulsification and chain extension reactions to form WPU dispersion [29,30]. The obtained WPU resin is nontoxic, nonflammable, nonpolluting, and is a good environmentally friendly material. The dried resin film has good mechanical properties, flexibility, gloss, and weather resistance [31].

However, to date, few studies have combined cellulose hydrogels and aerogels with WPU resins [32,33]. WPU dispersion has been introduced into the cellulose solution to form a cellulose/WPU composite network structure. The long-chain WPU resin is used as a toughening agent for cellulose hydrogels and aerogels, thereby improving the mechanical strength of the hydrogel and aerogel skeletons and reducing the weight loss during the process of water absorption and dehydration.

## 2. Materials and Methods

### 2.1. Materials

Microcrystalline cellulose (MC, average particle size 90 μm, Acros Organics, Geel, Belgium), carboxymethyl cellulose (CMC, molecular weight = 90,000, degree of substitution = 0.7, Acros Organics, extra pure), hydroxyethyl cellulose (HEC, molecular weight = 90,000, Polysciences, Inc., Warrington, PA, USA), sodium hydroxide (NaOH, Choneye Pure Chemicals, Taipei, Taiwan), urea (Choneye Pure Chemicals), epichlorohydrin (ECH, 99%, Acros Organics), polytetramethylene ether glycol (PTMG, molecular weight = 2000, Formosa Asahi Spandex Co., Ltd., Taipei, Taiwan), isophorone diisocyanate (IPDI, Tycoon Enterprises Ltd., Kaohsiung, Taiwan), dimethylolpropionic acid (DMPA, Acros Organic), dimethylacetamide (DMAc, Choneye Pure Chemicals), ethylenediamine (EDA, Choneye Pure Chemicals), and triethylamine (TEA, Tedia Company, Fairfield, OH, USA).

### 2.2. Preparation of Cellulose Solutions

The method of cellulose hydrogel preparation used here followed a previous study [34]. MC, CMC, and HEC were used as cellulose raw materials. Urea/NaOH/H_2_O and NaOH/H_2_O with weight ratios of 4/6/90 and 4/96, respectively, were used as dissolving reagents. The cellulose and dissolving agent were mixed in a weight ratio of 5/95 and stirred at 400 rpm for 60 min at room temperature, followed by ultrasonic shaking for 5 min to prepare MC hydrogel. The mixture was placed in a freezer set at −18 °C and frozen for 12 h. After thawing, the mixture was stirred, ultrasonically shaken, and frozen again. The above freezing–thawing was repeated 3 times. After 3 freezing–thawing cycles, homogeneous cellulose solution was obtained.

### 2.3. Synthesis of WPU

A prepolymerization process prepared WPU resin with PTMG, IPDI, DMPA, TEA, EDA, and water as the polyol, isocyanate, internal emulsifier, neutralizer, chain extender, and dispersed phase, respectively. Before synthesis, PTMG was placed in a 1000 mL four-neck glass reactor and dried in a vacuum oven to remove water. Then, DMPA and IPDI were added and reacted at 80 °C with a stirring speed of 200 rpm under a nitrogen atmosphere to undergo prepolymerization until the NCO content reached the theoretical value, which was measured according to ASTM-D2572 to form NCO-terminated prepolymers. At this stage, the reaction was conducted with molar ratios of 1.6/1 and 1.5/1 for NCO/OH and DMPA/PTMG, respectively. After cooling to 50 °C, TEA was added for neutralization for 30 min, followed by the addition of a calculated amount of deionized water dropwise under the stirring rate of 400 rpm. Finally, EDA was added to undergo chain extension reaction for 60 min, followed by continuous stirring at room temperature overnight to obtain a NH_2_-terminated WPU dispersion. The WPU dispersion was passed through a syringe filter with a 0.45 μm pore size (Millipore) to remove impurity. After filtering, the nonvolatile content, viscosity, pH value, and surface tension of the WPU dispersion were measured. The viscosity, pH value, and surface tension were measured with a Brookfield DV-E rotary viscometer, Suntex SP2100 (Suntex Instruments Co., Ltd., Taipei, Taiwan), and FACE CBVP-A3 (Kyowa Interface Science Co., Tokyo, Japan) at 25 ± 2 °C.

### 2.4. Preparation and Measurement of Cellulose/WPU Composite Hydrogel

ECH was used as the cross-linker to link two C6 alcohol groups in cellulose. The cellulose solution and ECH mixture with a molar ratio of epoxy group to hydroxyl group of 3/1 was stirred at room temperature for 20 min. Then, the calculated amount of WPU dispersion was added and stirred for 1 min to make the mixture uniform. The nonvolatile weight ratios of cellulose to WPU were set to 100/0, 95/5, 90/10, 85/15, and 80/20, and the formulation of composite hydrogels are shown in Table 1. The mixture was poured into a mold with an inner diameter of 35 mm × 30 mm × 20 mm It was sealed and placed in an oven set at 60 °C to form a cellulose/WPU composite hydrogel. The obtained hydrogel was weighed (Wi) and then immersed in flowing water to remove unreacted components and swell it sufficiently. The weights (Ws) after soaking for 24 h and 48 h were measured, and the swelling ratio (%) of the hydrogel was calculated as follows:Swelling ratio (%) = (Ws − Wi)/Wi × 100(1)

The compressive properties of the hydrogel were measured using a strength testing machine (Shimadzu EZ Tester, Shimadzu, Kyoto, Japan) at a load speed of 5 mm/min. Five repetitions were completed for each sample.

### 2.5. Manufacturing of Cellulose/WPU Composite Aerogel

We placed the cellulose/WPU composite hydrogel in a refrigerator at −20 °C to freeze the water into an ice crystal. The frozen sample was dehydrated in a freeze dryer (YAMATO; DC-55, Yamato Scientific Co., Ltd., Tokyo, Japan) for 4 days to form a dried cellulose/WPU composite aerogel. The gel fraction of the resin was calculated using the following formula:Gel fraction (%) = W_d1_/W_1_ × 100(2)
where W_1_ and W_d1_ are the weights of the hydrogel and aerogel, respectively.

### 2.6. Microstructure of Cellulose/WPU Composite Aerogel

The microstructure of aerogel was observed with a tabletop electric microscope (Hitachi TM-1000, Hitachi, Tokyo, Japan). The sample was fixed on the top of the sample stub with conductive double-sided tape, and an acceleration voltage of 15 kV was applied.

### 2.7. Water Absorption Capacity of Cellulose/WPU Composite Aerogel

The cellulose/WPU composite aerogel (W_d1_) was immersed into distilled water, taken out, and weighed (W_r1_) at 24 h and 48 h, and the reswelling ratio was calculated as follows:Reswelling ratio (%) = (W_r1_ − W_d1_)/W_d1_ × 100(3)

Then, the sample was dehydrated again (W_d2_) in a freeze dryer, and the weight loss during the immersion process was calculated as follows:Weight loss (%) = (W_d1_ − W_d2_)/W_d1_ × 100(4)

### 2.8. Thermal Degradation Properties of Cellulose/WPU Composite Aerogel

The thermal degradation behavior was determined by a Perkin-Elmer Pyris 1 thermogravimetric analyzer (TGA; Perkin–Elmer, Inc., Norwalk, CT, USA) in a N_2_ atmosphere at a heating rate of 10 °C/min from 50 °C to 600 °C.

### 2.9. Statistical Analyses

SPSS (Software version 20, SPSS Inc., Chicago, IL, USA) was used for statistical analysis. One-way analysis of variance (ANOVA) with Scheffe’s test was used to calculate the statistical significance of results. The results are marked as English letters in the table and the bar graph, with different letters indicating statistical differences. The *p*-value was 0.05.

## 3. Results and Discussion

### 3.1. Properties of WPU Dispersion and Cellulose Solution

An amine-terminated WPU resin was prepared by a prepolymerization process. PTMG, IPDI, DMPA, TEA, and EDA were used as the raw material of polyol, isocyanate, internal emulsifier, neutralizer, and chain extender, respectively. The cellulose/WPU composite hydrogel was prepared by mixing WPU dispersion and cellulose solution and adding ECH as a crosslinking agent. Table 2 shows the basic properties of WPU dispersion and cellulose solution. WPU dispersion is a low-viscosity, weakly alkaline dispersion with a viscosity of 6.78 cps, a pH of 7.4, and a surface tension of 40 dyne/cm.

Cellulose has no solubility in water and most organic solvents. The NaOH/urea solvent system can be employed to dissolve cellulose and thus vary the application [35]. However, CMC and HEC, the modified celluloses, are highly water-soluble celluloses [36,37]. As the three kinds of cellulose are dissolved by reagents containing NaOH, these cellulose solutions are alkaline liquids with a pH greater than 11. However, the viscosity and surface tension vary with the type of cellulose. CMC cellulose solution has higher viscosity and higher surface tension of 1378 cps and 66 dyne/cm, respectively, while the viscosities of MC and HEC cellulose solutions were 56 cps and 57 cps, and the surface tensions were 46 dyne/cm and 42 dyne/cm, respectively.

### 3.2. Appearance of Cellulose/WPU Composite Hydrogel

In this study, cellulose/WPU composite hydrogels were prepared by mixing cellulose solution and NH_2_-terminated WPU dispersion. In preliminary experiments, it was found that if the cellulose solution, WPU dispersion, and ECH were mixed simultaneously, the reactivity of ECH to the amine group of the WPU dispersion is greater than the reactivity to the hydroxyl group of the cellulose. This phenomenon caused the molecular weight of the WPU resin to increase rapidly, forming agglomerates and separating them from the mixed system. To overcome this problem, the cellulose solution and ECH were mixed first and then left to react at room temperature for 20 min before adding the WPU resin. Figure 1A shows the appearance of the composite hydrogels prepared from three kinds of cellulose with different cellulose/WPU weight ratios. The hydrogel made from pure CMC and HEC is a transparent colloid, while the MC is a translucent colloid. However, with the increase in the amount of WPU added, its appearance gradually changed from transparent to opaque white colloid.

### 3.3. Physical Properties of Cellulose/WPU Composite Hydrogel

Table 3 shows the apparent density and gel fraction of cellulose/WPU composite hydrogels prepared under different conditions. Due to the high water content of hydrogels, the apparent densities of cellulose/WPU composite hydrogels are similar to that of water, which is between 0.91 g/cm^3^ and 1.07 g/cm^3^. The gel fraction is the weight ratio of the hydrogel after dehydration to that before dehydration, and it can represent the weight percentage of the network structure in the hydrogel. The gel fraction of the hydrogel prepared with pure CMC is higher than those of pure MC and pure HEC. The introduction of WPU has no significant effect on the gel fraction of MC/WPU composite hydrogel, which is between 9.01% and 9.64%. However, after adding 5% WPU, the gel fraction of CMC/WPU composite hydrogel decreased significantly, but as the proportion of WPU increased, the gel fraction gradually increased. The gel fraction of hydrogel prepared by pure HEC is the lowest among the three celluloses, but the introduction of WPU can slightly increase the gel fraction.

Figure 2 shows the swelling ratios of cellulose/WPU composite hydrogel after being immersed in water for 24 h and 48 h. Figure 2A indicates no significant difference in the swelling ratio of MC/WPU composite hydrogels with different mixing ratios when immersed in water for 24 h. When the immersion time was 48 h, the swelling ratio of the MC hydrogel without WPU increased to 29%, while the MC/WPU composite hydrogels showed only a slight change. This shows that adding WPU to MC hydrogel reduces its water absorption capacity and makes it reach a swelling equilibrium within 24 h. Figure 2B indicates the swelling ratio of CMC/WPU composite hydrogels. It can be found that they have the highest swelling ratio among the three cellulose/WPU composite hydrogels. The swelling ratio of 48 h immersion is more than twice that of 24 h immersion, indicating that CMC/WPU composite hydrogels take longer to reach the swelling equilibrium. Figure 2C shows the swelling ratio of HEC/WPU composite hydrogels. The swelling ratio of HEC/WPU composite hydrogels is higher than that of MC/WPU. There is no significant difference in the swelling ratio of different HEC/WPU composite hydrogels after immersion in water for 24 h. Notably, the swelling ratio upon water absorption for 48 h is further increased. Among them, the swelling ratio of pure HEC hydrogel is slightly higher HEC/WPU composite hydrogels.

### 3.4. Compression Properties of Cellulose/WPU Composite Hydrogel

Figure 1B–D takes CMC-80/20 composite hydrogel as an example to show the change in appearance under external force. Figure 1B,C show the appearance of the composite hydrogel before and after applying external force. It can be seen that this composite hydrogel can withstand large compression deformation (40%) without breaking. Figure 1D shows that the composite hydrogel can return to its original shape after removing the external force. This result indicates that the cellulose/WPU composite hydrogel has good toughness and elastic recovery ability.

Figure 2D–F show the compression stress–strain curves of composite hydrogels prepared under different conditions. Figure 2D–F are cellulose/WPU composite hydrogels with different weight mixing ratios made by using MC, CMC, and HEC as raw materials, respectively. As can be seen, the type of cellulose is the most critical factor affecting the compressive stress of the composite hydrogel. The ones with MC as the cellulose have higher compressive stress, followed by HEC and CMC. However, composite hydrogels made of the same cellulose with other WPU additions ratios have similar stress–strain curves. Table 3 further shows that there is no significant correlation between the stress of the composite hydrogel and the amount of WPU added before the 20% strain occurs. While the strain reaches 40%, the corresponding stress appears irregular. This result shows that changing the added amount of WPU cannot effectively improve the mechanical strength of the hydrogel.

### 3.5. Microstructure of Cellulose/WPU Composite Aerogel

In this study, freeze-drying was used to remove water from a cellulose hydrogel with a volume of 35 mm × 30 mm × 20 mm to form a dry aerogel. Figure 3 shows the appearance of cellulose/WPU composite aerogels produced under different conditions. It can be seen that the dehydrated aerogels have a white porous structure. Comparing the three aerogels made purely from MC, CMC, and HEC, MC aerogel maintains a blocky appearance similar to its hydrogel. Chang et al. (2010) pointed out that CMC hydrogel has electrostatic repulsion due to the COO^−^ groups [20]. The effect promotes the CMC hydrogel to absorb more water, and the cross-linked network structure undergoes a greater degree of swelling and deformation after water absorption. The photo shows that the CMC aerogel has a larger volume but a more irregular appearance after dehydration. As mentioned above, the gel fraction of the hydrogel with HEC as the cellulose raw material is relatively low, which may cause more intense volume shrinkage during dehydration and formation of irregular aerogels.

We compare the appearance of composite aerogels made of different types of cellulose. As the proportion of WPU increases, the volume shrinkage of MC/WPU and HEC/WPU composite aerogels decreases, and a more complete block appearance structure can be maintained. This shows that the introduction of WPU can effectively preserve the pore structure of MC and HEC hydrogels and reduce structural shrinkage during the dehydration process. However, CMC/WPU composite aerogels with different WPU content present irregular shapes, and their volume shrinkage is greater than those of simple CMC aerogels.

Figure 4 shows the microstructure of cellulose aerogels and cellulose/WPU composite aerogels prepared with different conditions. Molecular chains of MC and CMC were dispersed homogeneously and derived from the porous morphology. The MC aerogel is a porous structure, and its pore walls are composed of a continuous thin resin film with many irregular perforations. The microstructure of MC/WPU composite aerogel is similar to that of MC aerogel, but the wall’s surface is relatively rough. This may be caused by the accumulation of some WPU resin on the wall surface. As mentioned above, HEC hydrogels undergo significant shrinkage when they are dehydrated to form aerogels. The microstructure also shows that the aerogel lacks an obvious pore structure. The network structure of the pore wall is bent and collapsed and stacked. Comparing the morphology of CMC aerogel and CMC/WPU composite aerogel, it is found that CMC/WPU composite aerogel has smaller pores and thicker walls. Chang et al. (2010) indicated that the high absorption of CMC hydrogel is related to its electrostatic effect [20]. The introduction of WPU reduces the electrostatic effect inside the CMC hydrogel, which reduces the amount of water it can hold and causes the pores of aerogels to become smaller.

### 3.6. Water Absorption Capacity of Cellulose/WPU Composite Hydrogel

Figure 5 shows the reswelling ratios of composite aerogels with different cellulose/WPU weight ratios at 24 h and 48 h. Figure 5A shows that the reswelling ratio of MC/WPU composite aerogels is greater than that of the pure MC aerogel, and the composite aerogel with an MC/WPU weight ratio of 90/10, showing the most significant water absorption capacity. Extending water immersion time from 24 h to 48 h, the reswelling ratio of MC aerogel was further increased, while the MC/WPU composite aerogels did not change significantly. This result shows that adding WPU can increase the water absorption capacity of MC aerogels and shorten the time required to reach swelling equilibrium.

Figure 5B shows the reswelling ratios of CMC/WPU composite aerogels after immersing in water for 24 h and 48 h. CMC/WPU composite aerogels have a higher reswelling ratio than MC/WPU and HEC/WPU composite aerogels. In addition, the reswelling ratio of CMC/WPU composite aerogels is significantly greater than that of pure CMC aerogels. This result shows that the introduction of WPU can increase the water absorption capacity of CMC aerogel. However, the reswelling ratio at 48 h is much higher than that at 24 h, indicating that CMC/WPU composite aerogel needs longer to reach the swelling equilibrium. Similar to the other two cellulose/WPU composite aerogels, the CMC/WPU aerogel also has the most significant water absorption capacity with a weight ratio of 90/10.

Figure 5C shows the reswelling ratios of HEC/WPU composite aerogels. As can be seen, the reswelling ratio of HEC/WPU composite aerogels is higher than that of MC/WPU composite aerogels. However, there are no statistically significant differences in the reswelling ratios of composite hydrogels with different HEC/WPU weight ratios. The reswelling ratios after 24 h and 48 h water immersion are about 1582~2382% and 1963–2430%, respectively. However, the reswelling ratio of HEC/WPU composite aerogel with a weight ratio of 90/10 is higher than those under other conditions. In summary, the cellulose/WPU composite aerogel with a weight ratio of 90/10 has the largest water absorption capacity. The water absorption capacity ranking is as follows: CMC/WPU, HEC/WPU, and MC/WPU. Figure 5D shows the weight loss of cellulose/WPU composite aerogels prepared with different conditions after immersion in water. The weight loss values of aerogels made solely with cellulose are HEC, CMC, and MC, in descending order. The weight loss of the cellulose/WPU composite aerogels is lower than that of the cellulose aerogel, and the weight loss tends to decrease as the amount of WPU increases. Among them, the weight loss of the MC/WPU composite aerogel with a weight ratio of 80/20 is reduced from 41% of MC aerogel to 7%, and the CMC/WPU and HEC/WPU composite aerogels decreased from 44% and 53% to 25% and 9%, respectively. A previous study indicated that weight loss of commercial cellulose foam is 55.4%, which is similar to that of HEC aerogel [38]. After immersion in water, the aerogels absorbed a lot of water, causing a large expansion in volume. The swelling behavior causes partial bond breakage and release from the structure, resulting in weight loss. This result indicates that the introduction of WPU resin can lead to a more insoluble system and strengthen the network structure of the cellulose aerogel and reduce weight loss when immersed in water. The low weight loss may be attributed to the crosslinking efficiency [39].

### 3.7. Thermal Resistance of Cellulose/WPU Composite Hydrogel

Figure 6A,B show the TGA and DTG curves of WPU resin and CMC/WPU composite aerogels with different weight ratios. According to the TG curves, the thermal degradation of WPU mainly occurs between 250 °C and 550 °C, and can be divided into three degradation stages, as shown in the DTG curve. The thermal degradation of CMC aerogel and CMC/WPU composite aerogels mainly occur at temperatures between 250 °C and 350 °C. Compared with CMC aerogel, the weight loss curves of CMC/WPU composite aerogels appear at higher temperatures at temperatures lower than 350 °C, indicating that the introduction of WPU effectively improves the thermal resistance of CMC.

Table 4 summarizes the analysis data. Comparing the thermal resistance of composite aerogels with different CMC/WPU weight ratios, the one with a weight ratio of 90/10 has the best heat resistance with an initial thermal degradation temperature (onset) of 272 °C and the fastest thermal degradation temperature (peak) of 314 °C. The weight loss of CMC aerogel tends to be flat after the temperature reaches 330 °C with a weight loss of about 61%. In contrast, the weight loss of CMC/WPU composite aerogels tends to be alleviated at temperature higher than 350 °C. However, CMC/WPU composite aerogels have weight losses (63–70%) greater than CMC aerogel (61%), which may be related to degradation of the soft segment of WPU.

Figure 6C,D show the TG and DTG curves of different types of cellulose/WPU composite aerogels. The TG curve shows that MC/WPU and HEC/WPU composite aerogels have better heat resistance than CMC/WPU composite aerogel at a temperature lower than 350 °C. In comparison, MC/WPU and CMC/WPU composite aerogels have better heat resistance for temperatures above 350 °C. DTG shows that CMC/WPU composite aerogel undergoes more intense thermal degradation at lower temperatures, while the thermal degradation of MC/WPU composite aerogel is more moderate.

## 4. Conclusions

This study successfully introduced WPU resin into cellulose hydrogel to prepare cellulose/WPU composite hydrogel and aerogel. CMC/WPU composite hydrogels have swelling ratios higher than those of MC/WPU and HEC/WPU. The compression test shows that the structure of cellulose-based materials is the most important factor affecting the compressive stress of the composite hydrogel. The ones with MC as a raw material have higher compressive stress, followed by HEC and CMC. Cellulose/WPU composite hydrogel has good toughness and elastic recovery ability. The introduction of WPU can effectively maintain the pore structure of MC and HEC hydrogels and reduce volume shrinkage during dehydration. The microstructure shows that MC/WPU composite aerogel is porous, while HEC/WPU composite aerogel lacks an obvious pore structure. The cellulose/WPU composite aerogel with a weight ratio of 90/10 has the largest water absorption capacity, and the water absorption capacity values of the three celluloses are ranked as follows: CMC/WPU, HEC/WPU, and MC/WPU. The introduction of WPU resin can strengthen the network structure of the cellulose aerogel and reduce weight loss when immersed in water. The thermal degradation of cellulose/WPU composite aerogels mainly occurs at a temperature between 250 °C and 350 °C. The heat resistance of CMC aerogels can be improved by the introduction of WPU, and, at 90/10, the weight ratio of CMC/WPU has the best heat resistance.

## Figures and Tables

**Figure 1 polymers-14-00204-f001:**
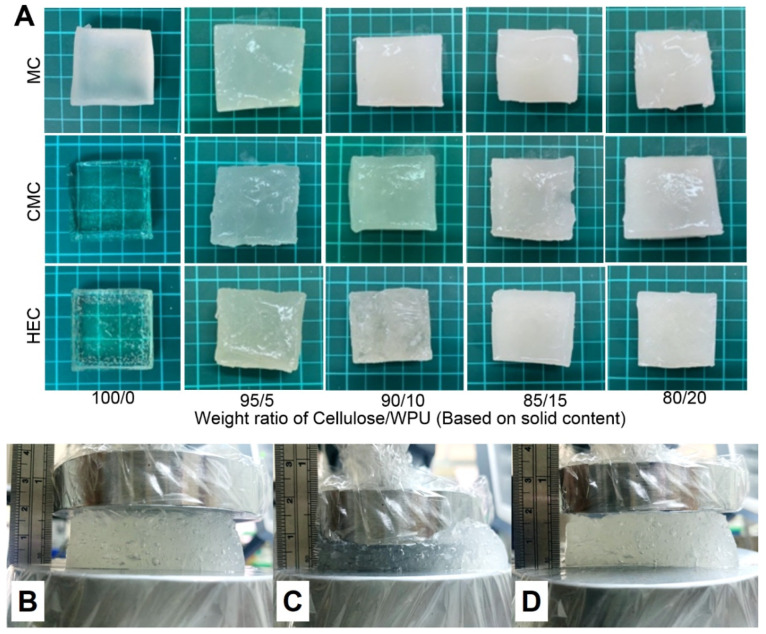
(**A**) Appearance of cellulose/WPU composite hydrogels. Changes in the appearance of composite hydrogel CMC-80/20 during compression (**B**); the appearance of the composite hydrogel before compression (**C**), after application of external force to strain 40%, and (**D**) after removal of the applied external force.

**Figure 2 polymers-14-00204-f002:**
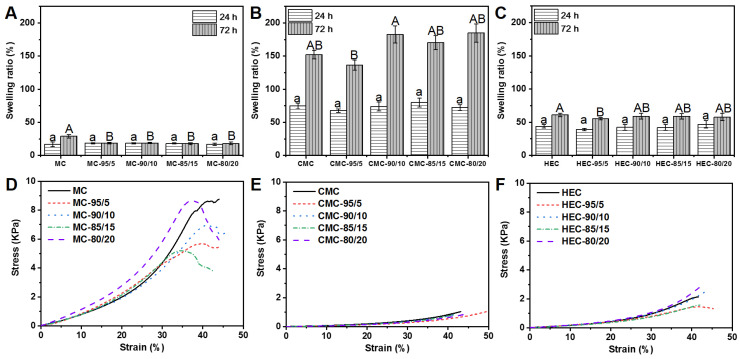
Swelling ratio (**A**–**C**) and stress-strain curves (**D**–**F**) of cellulose/WPU composite hydrogels. (**A**–**C**) Different letters indicate significant differences when using the same cellulose material at the 0.05 level obtained by Tukey’s test and analysis of variance (ANOVA). Different letters indicate significant differences between addition ratios of WPU in the same cellulose material at the 0.05 level obtained by Tukey’s test and analysis of variance (ANOVA). The letters of 24 h and 72 h are marked as “a and b” and “A and B”, respectively.

**Figure 3 polymers-14-00204-f003:**
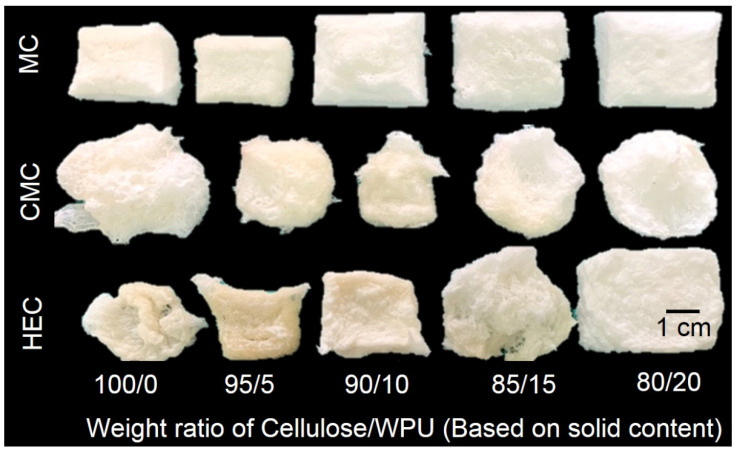
The appearance of cellulose/WPU composite aerogels.

**Figure 4 polymers-14-00204-f004:**
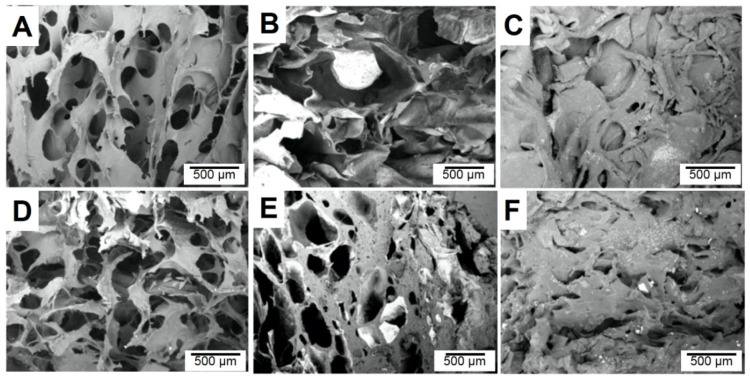
Microscope images of (**A**) MC; (**B**) CMC; (**C**) HEC cellulose and (**D**) MC-80/20; (**E**) CMC-80/20; (**F**) HEC-80/20 cellulose/WPU composite aerogels.

**Figure 5 polymers-14-00204-f005:**
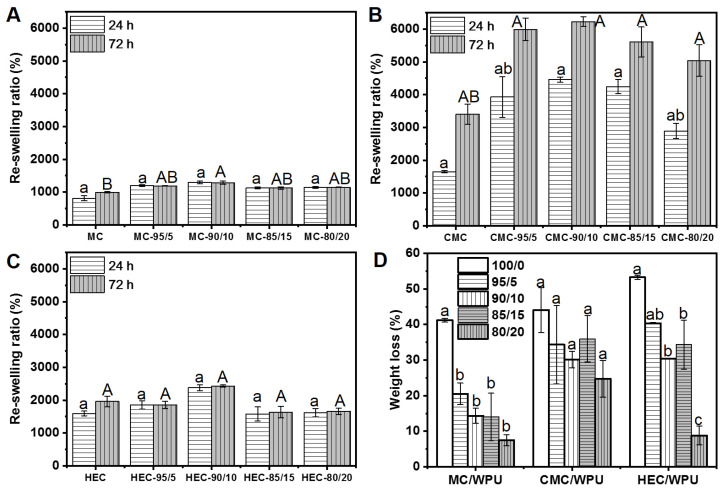
(**A**–**C**) Reswelling ratio of cellulose/WPU composite aerogels. (**D**) Weight loss of cellulose/WPU composite aerogel after water immersion. (**A**–**C**) Different letters indicate significant differences in reswelling time at the 0.05 level obtained by Tukey’s test and analysis of variance (ANOVA). Different letters indicate significant differences between addition ratios of WPU in the same cellulose material at the 0.05 level obtained by Tukey’s test and analysis of variance (ANOVA). The letters of 24 h and 72 h are marked as “a and b” and “A and B”, respectively.

**Figure 6 polymers-14-00204-f006:**
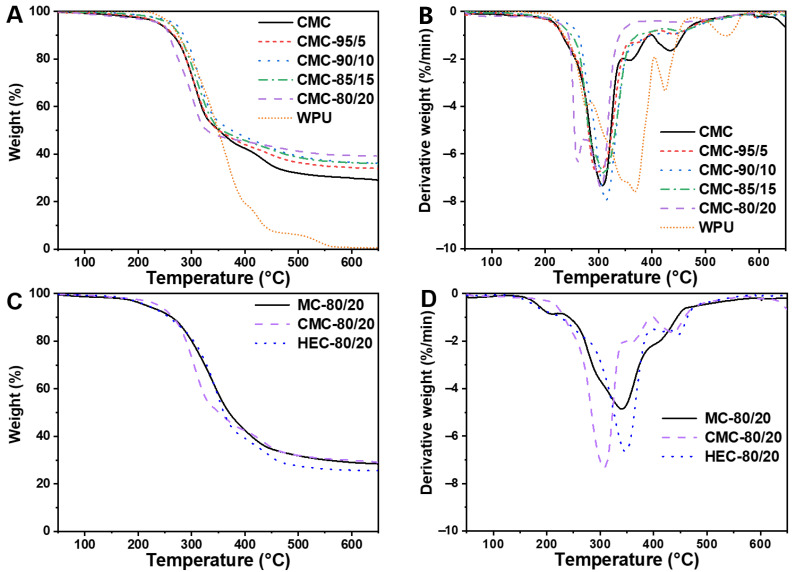
TGA curve of CMC/WPU composite aerogel with different weight ratios; (**A**) TG, (**B**) DTG. TGA curve of cellulose/WPU composite aerogel with different celluloses; (**C**) TG, (**D**) DTG.

**Table 1 polymers-14-00204-t001:** The formulation of cellulose/WPU composite hydrogels (weight by parts).

Sample Code	Microcrystalline Cellulose	Carboxymethyl Cellulose	Hydroxyethyl Cellulose	Waterborne Polyurethane
MC	100	-	-	-
MC-95/5	95	-	-	5
MC-90/10	90	-	-	10
MC-85/15	85	-	-	15
MC-80/20	80	-	-	20
CMC	-	100	-	-
CMC-95/5	-	95	-	5
CMC-90/10	-	90	-	10
CMC-85/15	-	85	-	15
CMC-80/20	-	80	-	20
HEC	-	-	100	-
HEC-95/5	-	-	95	5
HEC-90/10	-	-	90	10
HEC-85/15	-	-	85	15
HEC-80/20	-	-	80	20

**Table 2 polymers-14-00204-t002:** Properties of WPU dispersion and cellulose solution.

Sample	Nonvolatile (%)	pH	Viscosity (cps)	Surface Tension (dyne/cm)
WPU	30.0	6.8	7	40
MC	8.7	12.1	56	46
CMC	8.7	11.3	1378	66
HEC	8.7	11.1	57	42

**Table 3 polymers-14-00204-t003:** Apparent density, gel fraction, and compressive properties of cellulose/WPU composite hydrogels.

Sample Code ^1^	Apparent Density (g/cm^3^)	Gel Fraction (%)	Stress (kPa)			
			5% Strain	10% Strain	20% Strain	40% Strain
MC	0.99 ± 0.02 ^a,2^	9.50 ± 0.20 ^a^	0.37 ± 0.01 ^a,2^	0.77 ± 0.02 ^a^	1.90 ± 0.11 ^a^	9.14 ± 0.89 ^a^
MC-95/5	0.96 ± 0.03 ^a^	9.26 ± 0.31 ^a^	0.26 ± 0.05 ^a^	0.75 ± 0.10 ^a^	2.00 ± 0.23 ^a^	5.63 ± 0.28 ^bc^
MC-90/10	0.94 ± 0.03 ^a^	9.01 ± 1.02 ^a^	0.33 ± 0.07 ^a^	0.74 ± 0.17 ^a^	1.94 ± 0.29 ^a^	7.03 ± 1.61 ^abc^
MC-85/15	0.95 ± 0.04 ^a^	9.64 ± 0.30 ^a^	0.33 ± 0.04 ^a^	0.76 ± 0.04 ^a^	2.06 ± 0.11 ^a^	4.37 ± 0.35 ^c^
MC-80/20	0.96 ± 0.04 ^a^	9.41 ± 0.17 ^a^	0.44 ± 0.11 ^a^	1.00 ± 0.14 ^a^	2.58 ± 0.18 ^a^	7.83 ± 0.18 ^ab^
CMC	1.07 ± 0.10 ^a’2^	10.75 ± 0.71 ^a’^	0.06 ± 0.00 ^a’^	0.10 ± 0.01 ^a’^	0.23 ± 0.02 ^a’^	0.88 ± 0.14 ^a’^
CMC-95/5	1.00 ± 0.02 ^a’^	8.68 ± 0.37 ^c’^	0.04 ± 0.00 ^bc’^	0.07 ± 0.00 ^ab’^	0.17 ± 0.02 ^a’^	0.76 ± 0.20 ^a’^
CMC-90/10	0.96 ± 0.05 ^a’^	8.72 ± 0.80 ^c’^	0.05 ± 0.01 ^ab’^	0.09 ± 0.01 ^bc’^	0.22 ± 0.04 ^a’^	0.85 ± 0.14 ^a’^
CMC-85/15	0.96 ± 0.05 ^a’^	9.52 ± 0.20 ^bc’^	0.04 ± 0.01 ^bc’^	0.08 ± 0.01 ^abc’^	0.17 ± 0.03 ^a’^	0.82 ± 0.19 ^a’^
CMC-80/20	0.94 ± 0.02 ^a’^	10.63 ± 0.79 ^ab’^	0.04 ± 0.00 ^c’^	0.07 ± 0.00 ^a’^	0.16 ± 0.00 ^a’^	0.62 ± 0.04 ^a’^
HEC	0.91 ± 0.03 ^A2^	7.38 ± 0.37 ^B^	0.09 ± 0.01 ^A^	0.17 ± 0.01 ^A^	0.41 ± 0.03 ^A^	1.85 ± 0.30 ^BC^
HEC-95/5	1.02 ± 0.07 ^A^	8.24 ± 0.43 ^AB^	0.11 ± 0.00 ^A^	0.20 ± 0.01 ^A^	0.45 ± 0.02 ^A^	1.47 ± 0.05 ^C^
HEC-90/10	1.01 ± 0.04 ^A^	7.73 ± 0.35 ^AB^	0.12 ± 0.01 ^A^	0.21 ± 0.01 ^A^	0.50 ± 0.03 ^A^	2.24 ± 0.15 ^AB^
HEC-85/15	1.01 ± 0.04 ^A^	8.80 ± 0.56 ^A^	0.08 ± 0.00 ^A^	0.16 ± 0.00 ^A^	0.38 ± 0.00 ^A^	1.38 ± 0.10 ^C^
HEC-80/20	1.01 ± 0.05 ^A^	8.40 ± 0.15 ^A^	0.10 ± 0.02 ^A^	0.19 ± 0.02 ^A^	0.48 ± 0.02 ^A^	2.31 ± 0.14 ^A^

^1^ The number indicates the weight ratio of the nonvolatile content of cellulose to WPU. ^2^ Different letters indicate significant differences between addition ratios of WPU in the same cellulose material at the 0.05 level obtained by Tukey’s test and analysis of variance (ANOVA). The letters of MC, CMC, and HEC-based hydrogels are marked as “a, b and c”, “a’, b’ and c’”, “A, B and C”, respectively.

**Table 4 polymers-14-00204-t004:** TGA analysis parameters of cellulose/WPU composite aerogel with different weight ratios.

Sample Code	Onset ^1^ (°C)	Peak ^1^ (°C)	End ^1^ (°C)	Weight Loss (%)
WPU	247	367	398	99
CMC	241	305	331	61
CMC-95/5	263	311	350	65
CMC-90/10	272	314	353	63
CMC-85/15	260	296	340	66
CMC-80/20	252	307	482	70
MC-80/20	274	340	407	71
HEC-80/20	273	347	379	74

^1^ Onset, peak and end are the initial thermal degradation, the fastest thermal degradation, and the termination thermal degradation temperature, respectively.

## Data Availability

The data presented in this study are available on request from the corresponding author.

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
