# Peer review of "Bio-Based Hydrogel and Aerogel Composites Prepared by Combining Cellulose Solutions and Waterborne Polyurethane"

_polymers, 2022, doi:10.3390/polym14010204_

Round 1
Reviewer 1 Report
1) The authors use abbreviations in the abstract that are later used in the article without being mentioned for the first time in the text. Authors should cite the full name the first time a compound is mentioned in the article excluding the abstract.
2) At what temperature are cellulose solutions prepared with dissolving reagents?
3) Is the synthesis of WPU performed without any final purification?
4) Have the authors not evaluated the strength of the interface between cellulose and WPU?
5) Have the authors not characterized the aerogels and hydrogels produced?
6) Have the authors determined the morphological differences between MC and CMC and how do these differences influence the characteristics of the resulting composite?
Author Response
Response to Reviewer 1 Comments
Dear Reviewer,
Attached please find the manuscript entitled “Bio-based hydrogel and aerogel composites prepared by combining cellulose solutions and waterborne polyurethane”, by Ling-Jie Huang, Wen-Jau Lee, Yi-Chun Chen.
Enclosed please find the referenced manuscript (polymers-1496479). The comments from reviewers are gratefully acknowledged. The content has been modified followed by the reviewer’s requirement. Our responses and the revised parts (in red) in the manuscript were described in responses and revised manuscript. Our responses and the revised parts in the manuscript were described as followed:
1) The authors use abbreviations in the abstract that are later used in the article without being mentioned for the first time in the text. Authors should cite the full name the first time a compound is mentioned in the article excluding the abstract.
Response: Thank you for the kind suggestion. The full names and abbreviations were amended in the revised version. (Line 47, 53-54, 58, and 80)
2) At what temperature are cellulose solutions prepared with dissolving reagents?
Response: The cellulose and dissolving agent were mixed at room temperature. The information was added in the revised version (Line 106-108).
3) Is the synthesis of WPU performed without any final purification?
Response: WPU dispersion was passed through a syringe filter with a 0.45 μm pore size (Millipore) to remove impurity. After filtering, the non-volatile content, viscosity, pH value, and surface tension of the WPU dispersion were measured. The information was added in the revised version (Line 126-129).
4) Have the authors not evaluated the strength of the interface between cellulose and WPU?
Response: Thank you for the kind suggestion. Microcrystalline cellulose, carboxymethyl cellulose, and hydroxyethyl cellulose were first dissolved into a solution during the experiment. WPU dispersion and cellulose-based solution were mixed to form a homogeneous mixture in a single phase. The mixture has no clear interface to measure the strength between cellulose and WPU. Therefore, we can’t evaluate the strength of the interface.
5) Have the authors not characterized the aerogels and hydrogels produced?
Response: The swelling ratio and compressive properties had been tested for the hydrogel, and the gel fraction, microstructure, water absorption capacity (re-swelling ratio and weight loss), and thermal degradation properties were measured for aerogel.
6) Have the authors determined the morphological differences between MC and CMC and how do these differences influence the characteristics of the resulting composite?
Response: Original MC and CMC both are solid powder, which are aggregates. In the experiment, MC and CMC were first dissolved to disperse the molecular chains and then mixed with the WPU suspension. Therefore, the original solid particle morphology no longer exists in the hydrogel. The morphology of MC and CMC aerogels are similar to their composite in Figure 4. Molecular chains of MC and CMC were dispersed homogeneously and from the porous morphology. The information was added in the revised version (Line 303-304).
Reviewer 2 Report
The manuscript describes an interesting set of hydrogel/aerogel composites combining varied cellulose grades with waterborne polyurethane. The methods to prepare the materials seem careful, and the studies are quite systematic. The manuscript would be improved if the authors discussed the underlying mechanisms behind the obtained results more extensively. The results are now mainly presented as they are, with few attempts to interpret the meaning of the results or link different measurements to one another. The discussion related to Fig. 4 is a welcome exception to this. Moreover, it would be interesting to see the compression properties of the aerogels in addition to those of the hydrogels.
There are a few points where the text lacks clarity. These are indicated in below:
- Abstract: The idea of the opening sentence, “The hydrogel prepared from cellulose combines the advantages of cellulose and gel materials, and the aerogel formed after removing the water”, is unclear. Please modify.
- Abstract, last sentence: “…is the best ratio to make the cellulose/WPU composite aerogel have better water swelling capacity and heat resistance”; do you mean “…is the best ratio to make the cellulose/WPU composite aerogel with highest water swelling capacity and heat resistance”?
- Page 1, 2nd paragraph: ”… the characteristics of repeated water absorption and dehydration”; should there be “potential” instead of “characteristics”?
- Page 1, 2nd paragraph: “ …the cellulose gel appears the weight loss”; should there be “experiences” instead of “appears”?
- Page 1, 2nd paragraph: “…to improve this phenomenon”; write “property” instead of “phenomenon”.
- Page 2, 3rd paragraph: “…there are still few literatures”; write “papers” instead of “literatures”.
- Page 2, Section 2.1 Materials: Introduce again full names of the materials here even though these names have already been given in the abstract.
- Page 3, Section 2.2: What was the reason for repeating the freezing-thawing cycle for three times?
- Page 5, Section 3.1: ”Cellulose has low solubility in water and most organic solvents. A specific solvent system such as NaOH/urea needs to be employed to vary cellulose application”; these sentences are not clear, perhaps the highlighted words are wrong.
- Table 3: Please give a clearer explanation for the letters used in the statistical analysis.
- Page 10, Fig. 4 caption lacks the description of the different cases A-F.
- Page 11, Fig. 5D and related text: Please discuss mechanisms of the weight loss after water immersion.
Author Response
Response to Reviewer Comments
Dear Reviewer,
Attached please find the manuscript entitled “Bio-based hydrogel and aerogel composites prepared by combining cellulose solutions and waterborne polyurethane”, by Ling-Jie Huang, Wen-Jau Lee, Yi-Chun Chen.
Enclosed please find the referenced manuscript (polymers-1496479). The comments from reviewers are gratefully acknowledged. The content has been modified followed by the reviewer’s requirement. Our responses and the revised parts (in red) in the manuscript were described in responses and revised manuscript. Our responses and the revised parts in the manuscript were described as followed:
The manuscript describes an interesting set of hydrogel/aerogel composites combining varied cellulose grades with waterborne polyurethane. The methods to prepare the materials seem careful, and the studies are quite systematic. The manuscript would be improved if the authors discussed the underlying mechanisms behind the obtained results more extensively. The results are now mainly presented as they are, with few attempts to interpret the meaning of the results or link different measurements to one another. The discussion related to Fig. 4 is a welcome exception to this. Moreover, it would be interesting to see the compression properties of the aerogels in addition to those of the hydrogels.
There are a few points where the text lacks clarity. These are indicated in below:
- Abstract: The idea of the opening sentence, “The hydrogel prepared from cellulose combines the advantages of cellulose and gel materials, and the aerogel formed after removing the water”, is unclear. Please modify.
Response: Thank you for the kind suggestion. The sentence is modified to “The hydrogel composite prepared from of cellulose-based materials and the other gel material, which can combine the advantages of them. The aerogel formed after removing the water in the hydrogel can maintain a network structure and form a porous material. Both materials have high application potential.” The sentence was amended in the revised version. (Line 9-12)
- Abstract, last sentence: “…is the best ratio to make the cellulose/WPU composite aerogel have better water swelling capacity and heat resistance”; do you mean “…is the best ratio to make the cellulose/WPU composite aerogel with highest water swelling capacity and heat resistance”?
Response: Yes. The sentence was amended in the revised version. (Line 21)
- Page 1, 2ndparagraph: ”… the characteristics of repeated water absorption and dehydration”; should there be “potential” instead of “characteristics”?
Response: The word was amended in the revised version. (Line 38)
- Page 1, 2ndparagraph: “ …the cellulose gel appears the weight loss”; should there be “experiences” instead of “appears”?
Response: The word was amended in the revised version. (Line 42)
- Page 1, 2ndparagraph: “…to improve this phenomenon”; write “property” instead of “phenomenon”.
Response: The word was amended in the revised version. (Line 44)
- Page 2, 3rdparagraph: “…there are still few literatures”; write “papers” instead of “literatures”.
Response: The word was amended in the revised version. (Line 87)
- Page 2, Section 2.1 Materials: Introduce again full names of the materials here even though these names have already been given in the abstract.
Response: The full names and abbreviations were amended in the revised version. (Line 95-97)
- Page 3, Section 2.2: What was the reason for repeating the freezing-thawing cycle for three times?
Response: The method of MC hydrogel preparation followed by a previous study [32]. The mixture was placed in a refrigerator at -18°C and frozen for 12 h. After thawing, the mixture was stirred, ultrasonically shaken, frozen again, and repeated the above freezing-thawing 3 times. After freezing-thawing cycles 3 times, a homogeneous MC solution was obtained. The information was added in the revised version. (Line 108-112)
- Page 5, Section 3.1: ”Cellulose has low solubility in water and most organic solvents. A specific solvent system such as NaOH/urea needs to be employed to vary cellulose application”; these sentences are not clear, perhaps the highlighted words are wrong.
Response: Cellulose has no solubility in water and most organic solvents. NaOH/urea solvent system can be employed to dissolve cellulose to vary the application. The sentences were amended in the revised version. (Line 183-185)
- Table 3: Please give a clearer explanation for the letters used in the statistical analysis.
Response: Different letters indicate significant differences in using the same cellulose material at 0.05 level obtained by Tukey’s test and analysis of variance (ANOVA). The explanation was amended in the revised version. (Line 233-234)
- Page 10, Fig. 4 caption lacks the description of the different cases A-F.
Response: Thank you for the kind suggestion. Microscope images of (A) MC; (B) CMC; (C) HEC cellulose and (D) MC-80/20; (E) CMC-80/20; (F) HEC-80/20 cellulose/WPU composite aerogels. The description was amended in the revised version. (Line 319-320)
- Page 11, Fig. 5D and related text: Please discuss mechanisms of the weight loss after water immersion.
Response: Thank you for the kind suggestion. The previous study indicated that weight loss of commercial cellulose foam is 55.4%, which is similar to the HEC aerogel [36]. After immersing in water, the aerogels absorbed a lot of water and cause a large expansion in volume. The swelling behavior causes partial bond breakage and release from the structure, resulting in weight loss. This result indicates that the introduction of WPU resin can lead to a more insoluble system and strengthen the network structure of the cellulose aerogel and reduce weight loss when immersed in water. The low weight loss may be attributed to the crosslinking efficiency [37]. The discussion was added in the revised version. (Line 355-362)
